# Gelatin Films Modified with Acidic and Polyelectrolyte Polymers—Material Selection for Soft Gastroresistant Capsules

**DOI:** 10.3390/polym11020338

**Published:** 2019-02-15

**Authors:** Bartosz Maciejewski, Małgorzata Sznitowska

**Affiliations:** Department of Pharmaceutical Technology, Medical University of Gdańsk, Hallera 107, 80-416 Gdańsk, Poland; b.maciejewski@gumed.edu.pl

**Keywords:** gelatin, polyelectrolyte, gastroresistant polymers, disintegration, rheology

## Abstract

The following investigation comprised the formation of acid-resistant gelatin-based films, intended for future use in soft-capsule technology. Such film compositions were obtained by including nonionized forms of acid-insoluble polymers in a gelatin-based film-forming mixture. The selected films were additionally modified with small amounts of anionic polysaccharides that have potential to interact with gelatin, forming polyelectrolyte complexes. The obtained film compositions were subjected to, e.g., disintegration tests, adhesiveness tests, differential scanning calorimetry (DSC), and a transparency study. As a result of the performed study, some commercial enteric polymers (acrylates), as well as cellulose acetate phthalate, were selected as components that have the ability to coalesce and form a continuous phase within a gelatin film. The use of a small amount (1.5%) of additional gelling polymers improved the rheological characteristics and adhesive properties of the obtained films, with ί-carrageenan and gellan gum appearing to be the most beneficial.

## 1. Introduction

Gelatin is a very well-known natural polymer with polyelectrolyte properties, used in many applications associated with the food industry or pharmaceutical technologies, utilized as a thickener, gelling agent, and film-forming substance [1,2,3,4]. The characteristic feature of gelatins is the helix–coil transition of the molecules that occurs in gelatin solutions. Transition is dependent on the temperature of the system, and allows solidification of the gelatin solution by cooling down to temperatures around 25 °C. This property is utilized, e.g., in pharmaceutical-capsule technologies, especially in soft-capsule formation, where its unique ability to solidify on cooling is required to form capsule shells with the rotary die method, while liquefaction of the film surface on heating allows for providing the tight sealing of the capsule during the same process [5].

Gelatin-capsule shells do not naturally provide the modification of the dissolution profile of active ingredients. For such purposes, modification of the shell composition is necessary, either by changing the physiochemical characteristics of the gelatin or by combining gelatin with other polymers. Both methods are very challenging, and only a few products have been successfully manufactured, all with a coating technique. A significant modification of gelatin-capsule shells is necessary for achieving an enteric dosage form: insoluble at acidic pH, and soluble in a neutral (intestinal) environment. The traditional way to produce gastroresistant gelatin shells was to crosslink the chains in gelatin molecules with formaldehyde, but, due to toxicological reasons, use of this technique was discontinued [5]. Attempts of crosslinking the gelatin with enzymes or less toxic aldehydes have also been reported [4,6,7,8]. 

Although acid-resistant polymers, such as methacrylic acid–methyl acrylate copolymers (i.e., Eudragit^®^ L or S) are widely used for coating tablets, such coating is a more problematic process in the case of gelatin capsules [9,10]. The most desired would be a technology employing gelatin films incorporating acid-resistant polymers. Modification of shell composition in particular requires an understanding of the interactions between gelatin and additive polymers. There is, however, a lack of published experimental data that could support conceiving principles for such modifications. 

In our previous work, we successfully obtained gelatin-based films by including cellulose acetate phthalate pseudolatex dispersion in the film composition [11,12]. The films were investigated using various methods, i.e., scanning electron microscopy, confocal laser scanning microscopy, energy dispersive X-ray spectroscopy and Fourier transform infrared spectroscopy [12]. However, an attempt to identify the nature of the interaction between gelatin and cellulose acetate phthalate (CAP) particles was unsuccessful, so it is still unclear how important chemical interactions are between functional groups of gelatin’s amino acids and the groups in added polymers, in modifying the physical properties of gelatin-based films. 

Gelatin, as a polyelectrolyte, can interact with a wide range of compounds. At a pH below its isoelectric point (depending on the type of gelatin, e.g., for porcine gelatin pI, it is around 9 [3]), gelatin becomes partially positively charged, which enables the possibility to form bonds with substances that exhibit a negative charge, e.g., natural polysaccharide gums, where the negative charge usually arises from the presence of carboxyl or sulphate moieties. The formed polyelectrolyte complexes can appear in a soluble state, as a precipitate, or as a coacervate [13,14]. Besides interactions based on the net charge of particles, gelatin can also form covalent links with other substances [15,16,17], but the usefulness of covalent modification of gelatin appears to be less pronounced. In the following study, two types of polymers have been used: (1) ionic compounds, such as carrageenan, gellan gum, or xanthan; and (2) acid-insoluble polymers in nonionic forms, such as methacrylates, CAP, hypromellose phthalate (HPMCP), and other acid-insoluble cellulose derivatives. It is worth mentioning that acid-resistant polymers become ionized and soluble at neutral pH, which in vivo results in the dissolution of the modified capsule shell in the intestine.

The aim of this study is to assess the changes in the properties of films obtained by mixing gelatin and various acid-insoluble polymers of different commercial types, as well as to identify the influence of polysaccharides on the functional and mechanical properties of such film-forming mixtures and resulting films. The following study can lead to a better understanding of the conditions underlying the choice of modifying components in gelatin films that can secure not only gastroresistance, but also the required mechanical properties in the manufacturing process of soft capsules. Differential scanning calorimetry (DSC) was applied with the aim to provide preliminary conclusions on the possible chemical interactions between the polymers in the proposed compositions.

## 2. Materials and Methods 

### 2.1. Materials

Gelatin (porcine, type A) was generously provided by Medana Pharma (Sieradz, Poland), Aquacoat CPD^®^ (FMC Biopolymer, Philadelphia, PA, USA), a 30% aqueous pseudolatex dispersion of cellulose acetate phthalate (CAP) and cellulose acetate propionate (CAP 482-0.5), cellulose acetate butyrate (CAB 381-0.5) (Eastman Chemical, Kingsport, TN, USA), hypromellose phthalate (HPMCP(HP-55), hypromellose acetate succinate (HPMCAS) (Aqoat AS-MF and AS-HF, Shin-Etsu, Tokyo, Japan) were gifts from IMCD Polska (Warsaw, Poland). Eudragit L30 D-55, Eudragit L100, and Acryl Eze II (Evonik, Essen, Germany) were obtained from Evonik. Opadry Enteric (Colorcon, Budapest, Hungary) was provided by Colorcon. Aquarius Control ENA (Ashland, Covington, KY, USA) was received from Ashland. Glycerol (99.5%) was purchased from Chempur (Piekary Śląskie, Poland), polyethylene glycol (PEG-400), sorbitol, ί-carrageenan, gellan, and xanthan were purchased from Sigma Aldrich (Saint Louis, MO, USA). The compositions of certain complex products are specified further in Table 1.

### 2.2. Film Composition and Preparation

All film compositions contained gelatin, acid-resistant additive, and plasticizer. The mass ratio of the acid-resistant component to gelatin was 1:3. The compositions contained glycerol as a plasticizer, in an amount of 0.46 per 1 g of total polymer amount. The first step of film preparation was mixing all components with water. The initial water content in all compositions was 40% *w*/*w*. The mixture was heated in a water bath to 80 °C while being stirred with a paddle stirrer at 40 rpm for 2 h. Afterward, it was deareated under vacuum. The hot film-forming mass was then casted on a glass plate, and the thickness of the film was aligned by sliding through a 2000 µm gap, using a Camag TLC plate coater (Camag, Muttenz, Switzerland). Afterward, the films were dried in an air dryer at room temperature for 90 min and finally equilibrated at room temperature/15–25% RH (relative humidity) for at least 24 h. This procedure resulted in dry films with 450–700 µm thickness.

Two sets of samples were prepared. The first set contained samples intended to evaluate the possibility of preparing the acid-resistant gelatin-based films by adding various enteric coating polymers and their commercial mixtures to the gelatin. The second set was prepared to evaluate the influence of added cogelling agents to the gastroresistant gelatin mixture on the final performance of the films. The compositions are shown in Table 1 and Table 2.

### 2.3. Visual Evaluation and Disintegration Test

The obtained films were evaluated in terms of uniformity and their ability to set at room temperature. Additionally, the films were assessed in terms of mechanical durability during separation from the base plate. The films that showed suitable setting behavior and mechanical resistance were subjected to a disintegration test. Additionally, photos of certain films were taken with an Opta-Tech X2000 optical microscope (Opta-Tech, Warsaw, Poland).

Disintegration of the films was evaluated using a pharmacopoeial (European Pharmacopoeia, Ph. Eur.) tablet disintegration tester (ED 2SAPOx, Electrolab, Mumbai, India). Film samples 1 × 1 cm were placed in steel mesh sinkers before being introduced to the apparatus baskets. The test was performed at 37 °C using Ph. Eur. media (900 mL) at pH 1.2 (Simulated Gastric Fluid, SGF), pH 4.5 (phosphate buffer), and pH 6.8 (phosphate buffer). The baskets with the samples were repeatedly raised and lowered in the immersion fluid at a frequency rate of 30 cycles per min. The endpoint of the test was defined when any discontinuities in the film sample were observed.

### 2.4. Rheology

The rheology of film-forming mixtures was investigated using Haake Viscotester 550 (Thermo Fischer Scientific, Waltham, MA, USA) equipped with an E100 cylinder probe and Haake RheoWin software (V. 4.30.0021, Thermo Fischer Scientific, Waltham, MA, USA). The hot samples (40 mL) were placed in a heated beaker at 80 °C, and the beaker was covered and left for 5 min to restore the gel structure of the sample. The test was performed at 80 °C with a shear rate increasing from 20 to 50 (1/s) in 10 steps, where at each step the shear rate remained constant for 30 s. Then, the shear rate was decreased from 50 to 20 (1/s) in the same manner. Measurements were performed in triplicate. 

### 2.5. Light Absorbance and Transparency

Selected compositions were investigated in terms of the absorption of certain wavelengths of visible light using a Jasco V-530 UV/Vis spectrometer (Jasco International, Tokyo, Japan) and Jasco Spectra Manager software (V. 1.53.00, Jasco, Easton, MD, USA). The film samples were vertically mounted to the distal wall of the cell holder. Air was a reference. The transparency of the film samples was assessed, calculated as a function of transmittance measured at 600 nm, using the following equation [18]:
$$F=\frac{-\log\left(T_{600}\right)}{x},$$
where *T*_600_ is transmittance at 600 nm wavelength; *x* is film thickness (mm).

### 2.6. Texture Analysis

The selected films were tested in terms of tensile strength (TS, the force measured at sample disruption divided by sample cross-section area), elongation at break (EAB%, elongation at sample disruption in relation to initial sample length), and Young’s modulus (E, defining the relationship between stress and strain in the linear elasticity regime of the deformation of the investigated samples). 

Prior to testing, the film samples were conditioned at room temperature and 15–25% RH for at least 24 h. Films were cut into strips (0.5 × 5 cm) and subjected to a uniaxial tensile test utilizing the TA.XT Plus texture analyzer with a 5 kg load cell (Stable Micro Systems, Godalming, UK) and Stable Micro Systems Exponent software. Films were tested in triplicates. The air temperature in the laboratory was 18.4 °C and relative humidity was 31% during the whole investigation. The thickness and moisture contents of the films were evaluated prior to the test.

### 2.7. Self-Adhesiveness Test

The adhesiveness test was designed and performed using the TA.XT Plus texture analyzer (Stable Micro Systems, Godalming, UK) and Stable Micro Systems Exponent software to test the feasibility of the modified films to be sealed with a conventional soft-capsule sealing method. Samples of nondried films (1 × 5 cm), i.e., directly after casting, were fused together on a distance of 1 cm by squeezing with a clip and placing at temperature 40 or 60 °C for 3 min. The fused films were cooled down to room temperature. The free ends of the fused films were placed in tensile grips of the analyzer. The experimental setup is shown in Figure 1. The force corresponding to the resistance of the fused films to separation was measured. During the test, laboratory conditions were controlled at 24 °C ± 0.5 °C /40% ± 5% RH.

### 2.8. Differential Scanning Calorimetry (DSC)

Thermal analysis of the films was conducted using Netzsch DSC 204F1 Phoenix equipment with Netzsch Proteus software (Netzsch, Selb, Germany). The film samples (5 mg) were placed in an aluminum pan and heated in a nitrogen atmosphere from 30 to 150 °C with a rate of 10 K/min in a single cycle (without a cooling phase), with the empty aluminum pans as a reference. The films containing gelatin and carrageenan were investigated in terms of detecting the formation of a polyelectrolyte complex. Moreover, films modified with Aquacoat CPD or Eudragit L (GA, GE) were investigated and compared soon after preparation and after storage for 10 months at room temperature/15–25% RH.

## 3. Results

### 3.1. Visual Evaluation and Disintegration Test

In the first step of the study, the combination of gelatin with acid-resistant polymers was investigated (Table 1). Only in some cases were pure polymers used (GHP, GP, GB, GS-MF, and GS-HF). In other formulations, commercial mixtures containing additives were used. Among them were “ready-to-use” products containing methacrylate polymers (GACR, GAQ) or PVAP (GOP). In some cases, polymers were utilized in the form of commercial aqueous dispersions (GA, GE). All prepared compositions were allowed to obtain film-forming masses that, after casting, solidified at room temperature and created elastic films. The appearance of the selected films is shown in Figure 2. In case of few compositions, however, issues associated with homogeneity were identified after casting on a glass plate.

Clear, uniform films were obtained when GA, GE, and GE100 compositions were casted. The films were transparent or slightly opalescent, elastic, and tear-resistant on separation from the base plate. Even though the acid-resistant polymers were added either in the form of “easy-to-use” aqueous dispersion (GA, GE) or as a powder (GE100), there were no signs of visible particles within the film structure. This phenomenon was observed in previous work [12] and can be explained by the low glass-transition temperature of the used polymers that leads to the fast coalescence of latex particles on heating. In the above-mentioned study, it was observed that probably no chemical interaction occurred between gelatin and the added acid-resistant polymer, and it was more likely that both polymers were present in the films as two separate but intermixed phases, which is sufficient to achieve the acid-resistant properties of the films. Uniform films were also obtained in the case of the GOP, GACR, and GAQ compositions, which contained “ready-to-use” enteric-coating mixtures. The films were, however, opaque due to the presence of colorants in the employed coating mixture.

Some films contained solid particles or particle agglomerates, visible in the film structure. Inhomogeneities were observed in the case of the following film compositions: GB (containing cellulose acetate butyrate), GP (containing cellulose acetate propionate), and GHP (containing hypromellose phthalate); all three formulations contained solid particles, incorporated uniformly in the cloudy film structure. Major inhomogeneity was observed in the case of GS-MF and GS-HF films, both containing HPMCAS polymer, added to the composition in the form of micronized powder. These films in some regions contained big lumps of agglomerated additive polymer, incorporated in a cloudy film with a rough surface. For the disintegration test, fragments without major agglomerates were cut out.

The current standards for the disintegration time of gastroresistant dosage forms state that the investigated sample should not disintegrate in 0.1 M HCl in 2 or 3 h (depending on the composition, but not less than 1 h), which should be followed by disintegration within 1 h at pH 6.8 (Ph. Eur. 9.0). The results of the film-disintegration time test are shown in Figure 3. The films were first tested at pH 1.2 in simulated gastric fluid without pepsin. The test duration was 3 h. 

Films GA, GE, GE100, and GOP passed the test. Due to the homogeneous appearance of the films, it can be expected that full coalescence of the additive polymer occurred, rendering the compositions acid-resistant according to the mechanism described before [12]. However, the GHP composition, containing HPMCP, was a surprise. Roughness of the film structure and visible solid particles may suggest that no coalescence occurred between the HPMCP particles. Notwithstanding, the GHP composition did not disintegrate at the investigated acidic conditions for 3 h. However, the residue after the disintegration test was very soft and jellified, which may suggest that the GHP film was visually not disintegrating in acid, but became permeable.

Compositions containing HPMCAS did not endure the acid phase and disintegrated within ca. 3 min (GS-MF) and 2.5 min (GS-HF). Similarly, compositions GB (containing CAB) and GP (containing GP) did not show resistance to disintegration at low pH, which proves that the acid-resistant polymer did not form a continuous phase, and its solid particles that dispersed in gelatin film structure were insufficient to modify disintegration profile of the film.

Disintegration of the GACR and GAQ films (both containing ready-to-use powder mixtures for enteric coating based on methacrylate polymers) was a surprising result. Although the acid-resistant polymer in these compositions was the same as in GE and GE100, and the smooth surface of films with high elasticity would rather suggest homogeneous film formation, these compositions disintegrated at pH 1.2 in ca. 8.5 and 13 min, respectively, while GE and GE100 films showed resistance to low pH. This indicates that not only the type of polymer, but also the presence of certain excipients, influences the formation of acid-resistant structures in the film under certain conditions.

Compositions that passed the disintegration test at pH 1.2 (GA, GE, GE100, GHP, GOP) were subjected to disintegration at pH 4.5, which was used to simulate an increase in pH in a fed stomach [19], and pH 6.8, which is generally used to simulate the pH of intestinal fluid. The objective was to select film compositions that are resistant to disintegration at low pH, but show fast disintegration or dissolution at intestinal pH and, preferably, do not disintegrate at pH 4.5. All tested compositions fulfilled the requirements of fast disintegration at pH 6.8, but only the GA composition (containing Aquacoat CPD, which is based on CAP) turned out resistant to pH 4.5, which confirmed previous studies on the subject [11]. The GA composition was then selected for further investigation.

### 3.2. Rheology

In the next step of the study, gastroresistant GA was modified by the addition of a secondary gelling agent such as carrageenan, gellan, and xanthan, as described in the Methods section (Table 2). In these types of polymers, one can expect that polyelectrolyte complexes with gelatin are formed [20,21,22], which may change some of the physical properties of the films. 

Figure 4 shows results of rheological measurements of five investigated film-forming mixtures: GEL (as reference), GA, GAC, GAG, and GAX. The compositions were investigated under isothermal conditions at 80 °C, which reflects the temperature of the gelatin mass prior to film casting during soft capsules manufacturing. Thermostated hot mixtures were investigated with a cylinder probe according to the test protocol described in the Methods section. Three subsequent measurements of the same sample were performed within 30 min (each run lasted 10 min). Such design of the experiment was aimed to better differentiate rheological properties of individual compositions and to detect changes caused by additives. In Figure 4, red curves refer to 1st run, and green and blue for 2nd and 3rd replicates, respectively.

There is a large difference between viscosity values measured for GEL and GA formulations. It is demonstrated that the viscosity of gelatin mass is much lower (more than two times) upon addition of CAP (GA composition), which can cause trouble in the industrial production of soft gelatin shells because too small viscosity of the casted mass creates risk of inability of forming films with uniform and proper thickness. However, a small amount of content of cogelling polymers in polysaccharide-modified mixtures (GAC, GAX, GAG) resulted in significant increase of the viscosity, which was similar to the viscosity of GEL mass or even higher. This can be considered an advantage in terms of handling of such film-forming masses.

The analysis of the viscosity and shear stress curves generated during three subsequent measurements also demonstrates significant difference between GEL and GA formulations. A very pronounced dilatant behavior of the GEL mass is observed, which can be probably explained by easy gelation of the mass in contact with air during mixing, when water evaporation from the hot mass also occurs. Moreover, in the GA and GAG modified mass, dilatancy is still visible, however to a much lesser extent. When carrageen or xanthan gum is added to GA, the mixtures are physically very stable and no changes in the curves generated in the subsequent three runs were observed (GAC and GAX). Therefore, besides other advantages of gelling system modifications, addition of cogelling agents to the formulation could result in better processability of GA-based compositions.

The obtained results suggest a stabilizing effect of a polysaccharide on GA mass, which is suspected to be based on an ionic interaction. According to the literature [3], the isoelectric point (pI) of gelatin is about 9.0. Xanthan gum, which is a polysaccharide polymer that can form complex hydrogels with gelatin based on electrostatic interaction when the pH of the mixture is around or below the pI of gelatin, and the temperature is above the coil–helix transitions of both materials [21]. Similarly, gellan and carrageenan are anionic polysaccharides that form mixed gel networks, stabilized by the electrostatic attraction of molecules [22,23]. Regardless of the polymer and other additives used, the pH was, in all compositions, in the range of 4–5, which was below the pI of gelatin. Since the ratio of gelatin and these additional polymers was high (ca. 34:1), the system was still rich in ionized functional groups originating from gelatin.

### 3.3. Differential Scanning Calorimetry

The DSC curves of compositions GA and GAC were investigated in search of changes of thermal effects attributed to interaction between gelatin and carrageenan. The results, however, did not provide clear confirmation of the interaction between gelatin and carrageenan in GAC films: only a slight peak shift from 62.2 to 61.4 °C and a small change in Δ*H* (from 0.7619 to 0.9064 mW/mg) were observed. As carrageenan content in the film is very little (the gelatin to carrageenan ratio is 100:3), classical DSC equipment may not be sensitive enough, and a micro-DSC technique could be more suitable for studying potential interactions [24].

DSC analyses were also conducted to assess the stability of GA films on storage. Figure 5 shows changes in the DSC curves of the GA composition upon storage for 10 months at room temperature and 15–25% RH. The change in Δ*H* from 0.7619 to 0.7588 mW/mg is not significant; however, a shift in peak temperature value from 62.2 to 64.2 °C cannot be ignored. Due to the fact that the functional properties of the films (resistance to acid, while fast dissolution at pH 6.8, evaluated with the disintegration test) were not impaired, it can be concluded that there was no major alteration to the film structure on storage. 

### 3.4. Physical Properties of Casted Enteric GA Films

#### 3.4.1. Visual, Optic and Mechanical Properties

The effect of added cogelling polymers on the physical properties of GA enteric films was studied. Only the GAC film was observed to be completely homogeneous, while the GAG film contained graininess (Figure 6). In comparison to the GEL film, both the GA and GAC compositions were significantly less transparent, which was demonstrated by the spectrophotometric measurements (Table 3). The less transparency of the GA samples in comparison to GEL can be attributed to the formation of a bicontinuous phase of mixed polymers [25]. The lowest transparency of the GAC film was caused by the presence of carrageenan, which forms turbid gels when interacting with gelatin, resulting in the formation of two intermixed gels, stabilized by electrostatic interaction. The result is consistent with results from other analyses in the article, as well as with previously published works [24,26,27,28].

All of the prepared compositions displayed properties that were sufficient for mechanical manipulations. Only the GAX film turned out to not be mechanically resistant enough, and it was very hard to separate it from the glass plate after casting without damage.

The mechanical analysis of the dry films (GEL, GA, GAC and GAG) was performed according to the protocol described in the Methods section. The mechanical analysis results are shown in Table 4.

The highest Young’s modulus was measured for the non-modified GEL film samples. Moreover, the composition showed the highest mechanical resistance to tearing, exceeding the maximum force (50 N) that could be applied by the equipment. 

The films with an added acid-resistant polymer (GA) were significantly more fragile than what was demonstrated by lower TS and Young’s modulus values. It was to be expected, however, because the gelatin content in the films was reduced by the addition of the acid-resistant polymer from 68.6% to 51.5%, and the CAP additive does not form gel structures that could replace the gelatin structures in the composition. The addition of carrageenan (in GAC) or gellan (in GAG) lowered the tensile-strength and Young’s modulus values of the GA composition, although the EAB was practically unchanged. The mechanical properties of the GA film were not improved by the addition of cogelling agents, although these polymers increased the TS of gelatin films, as reported by other authors [24]. On the other hand, there are published results that increased elasticity was also not observed in gelatin–gellan or gelatin–carrageenan films [24,29].

#### 3.4.2. Self-Adhesiveness Test

Figure 7 demonstrates the recorded force during the attempt to separate two sheets of the investigated films that were compressed at 60 °C. At this temperature, the films became soft and adhesive, and fusion was possible. Adhesion was strong because no separation of the sheets was observed under the test conditions, and the graphs in Figure 6 represent the elongation of the fused samples before final break. 

The lower temperature applied during the film-sheet fusion, i.e., 40 °C, was sufficient to create a permanent junction between the films only in the case of the GAG samples, while the GA and GAC films were separated during the test.

## 4. Discussion

The performed investigation aimed for better insight into phenomena that occur in gelatin-based films modified with the addition of acid-insoluble polymers, as well as recognizing whether further modifications could be introduced in the system by the addition of other polymers with a polyelectrolyte character.

The current pharmacopoeial (Eur. Ph. 9.0) standards for the disintegration time of gastroresistant dosage forms state that the investigated sample should not disintegrate in 0.1 M HCl within 2 or 3 h (depending on the composition, but not less than 1 h), which should be followed by disintegration within 1 h at pH 6.8. For good in vivo performance, it is desirable that the dosage form is also resistant to the higher pH, namely, pH 4.5, occasionally occurring in stomach.

The acid-resistant polymers that were combined with the gelatin have acidic properties, and can be grouped into acrylate polymers (in the GACR, GAQ, GE, and GE100compositions), cellulose derivatives, i.e., phthalates, acetates, succinates (in the GA, GHP, GB, GP, GS-MF, the GS-HF compositions), and also polyvinyl acetate phthalate (GOP formulation). None of the above-mentioned polymers is ionized, and they are insoluble in acidic environments. The pH measurement showed that their combinations with gelatin were characterized by similar pH values, as was determined for the non-modified gelatin solution (pH 4–5). In this pH range, ionic interactions of the additive polymers with gelatin are not likely. Therefore, the resistance of the modified gelatin films to the HCl solution may result from the continuous structure formed by the enteric polymer within the gelatin gel network. Such a structure, macroscopically identified as transparent films, was obtained with CAP in the GA composition (added in the form of latex dispersion), as well as methacrylates (Eudragit type) in the GE and GE100 compositions. As a result, such films were insoluble at pH 1.2 (Table 1). Besides the films mentioned above, no other formulation was identified to form transparent films. This observation proves that only particular acid-insoluble polymers combined with gelatin are able to form a film structure that is gastroresistant. There is no clear explanation why some of the enteric polymers, belonging to different chemical groups, did not create gastroresistant gelatin films. Our preliminary studies with CPD and Eudragits demonstrated that a gelatin–enteric polymer ratio of 3:1 is optimal, but different ratios were not studied for the films that failed to be gastroresistant. One of the possible reasons for unsuccessful film formation is the precipitation of some enteric polymers when they are mixed with gelatin. Indeed, in most of these compositions, visible insoluble particles of the enteric polymer were present, which proved that no continuous phases of such polymers were formed within the film structure. Only one polymer can be indicated (HPMCP) that allowed the formation of acid-insoluble films, in spite of the fact that undissolved particles were observed in the film (GHP composition). However, during the disintegration test in acid, these films were clearly transitioned to a hydrogel. Unfortunately, due to coloration, it was hard to determine whether the PVAP polymer (in the GOP composition) formed a suspension or a continuous structure in the film. Due to the fact that the mentioned film composition neither disintegrated nor dissolved in HCl, it can be suspected that, in this case, the enteric polymer formed a rather continuous phase within the gelatin structure.

The formation of a continuous phase of an acid-resistant polymer was achievable due to the coalescence of dispersed polymer particles. Coalescence in such a system can occur when the composition is subjected to temperatures above the glass-transition temperature (*T*_g_) of a particulate polymer. This temperature can be lowered by the presence of solvents and plasticizers in the composition. Particularly good results in the case of CAP and methacrylates can be attributed to such process. However, it appeared that the presence of insoluble stabilizers in the used polymer mixtures was disadvantageous—the methacrylates in the GACR and GAQ formulations contained titanium dioxide, talc, and silicates, and acid-resistant films were not obtained in these systems.

From all formulated films, only the GA composition remained intact, also at pH 4.5, which most probably resulted from the properties of CAP itself (pKa value ca. 5.28 [30]) than from the exceptional properties of the obtained structures.

In the manufacturing process of soft capsules, the mechanical properties of gelatin films are crucial. In the GA composition, sufficient tensile properties were observed, although a slight decrease in the values of certain parameters was measured (in relation to non-modified gelatin films). However, due to the significant lowering of mass viscosity by the addition of CAP dispersion, additional gelling polymers were investigated as viscosity enhancers and stabilizers. The selected polymers had polyelectrolyte properties, but, as opposed to gelatin, which is a protein, the polymers were polysaccharides, such as ί-carrageenan, gellan, and xanthan gum. It appeared that the addition of small amounts of such polymers (1.5%) noticeably improved viscosity and altered the rheological characteristics of the film-forming mass. On the other hand, it was determined that the addition of cogelling agents only slightly influenced the mechanical properties of the films: tensile strength and elasticity. 

Particularly beneficial was the addition of carrageenan and gellan gum. Gellan meaningfully improved the self-adhesiveness of the films, which determines their fusion during the soft-capsule formation process. 

In spite of the observed alteration of some physical properties of the compositions, the DSC study did not allow for detecting any interaction between the polyelectrolytes present in the composition. The DSC technique could possibly be useful in tracking the changes within films in storage, as a small shift in peak temperature was observed in the stored films. The factors impairing the stability of gelatin-based films can be either structural water content (leading to potential hydrolysis or microbiological instability), or the tendency of gelatin to form crosslinks between its chains [6]. Because the functional properties of the stored films remained unchanged (data not shown), it is worth exploring how useful DSC analysis can be as an early indicator of the structural changes in gastroresistant gelatin polymers.

## 5. Conclusions

As a result of the performed study, a set of enteric polymers was selected as components that have the ability to coalesce and form a continuous phase within a gelatin film prepared in high-temperature conditions. The selected polymers comprised methacrylate copolymers (Eudragit type), CAP (in form of latex dispersion), and PVAP. No relation was observed between the chemical nature of the enteric components and their ability to form with gelatin acid-resistant films, but the formation of such films was most likely caused by the uninterrupted coalescence of dispersed polymer particles, resulting in the films’ homogeneous morphology. Interestingly, this process depended on the additives present in the commercial mixtures of the enteric polymers, and some of these products were unsuitable despite the same type of the present enteric polymers. The use of a small amount of additional gelling polymers with polyelectrolyte properties (polysaccharides) improved the rheological characteristics and adhesive properties of the obtained films, wherein the addition of ί-carrageenan or gellan gum appeared to be the most beneficial. Due to certain issues with film homogeneity regarding gellan (GAG) compositions, the carrageenan (GAC) composition appeared to be the most suitable for further development.

## Figures and Tables

**Figure 1 polymers-11-00338-f001:**
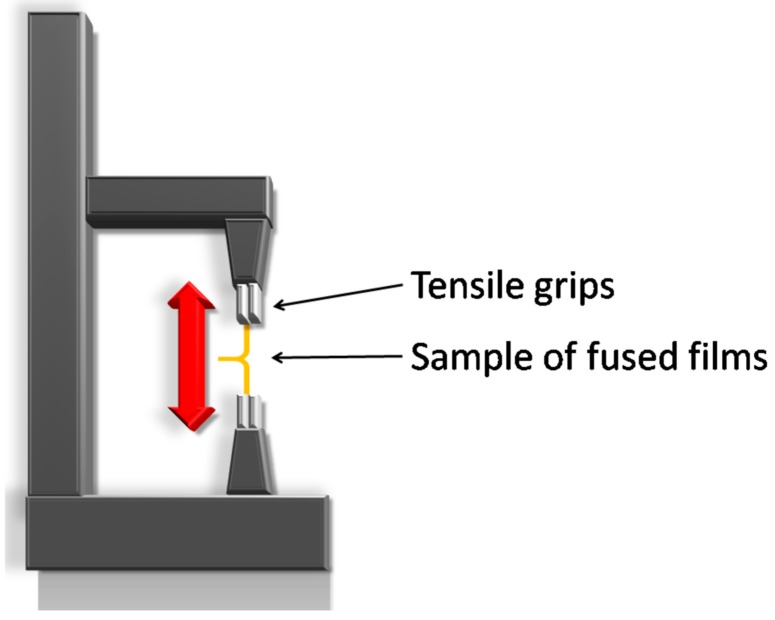
Schematic presentation of the self-adhesiveness test setup.

**Figure 2 polymers-11-00338-f002:**
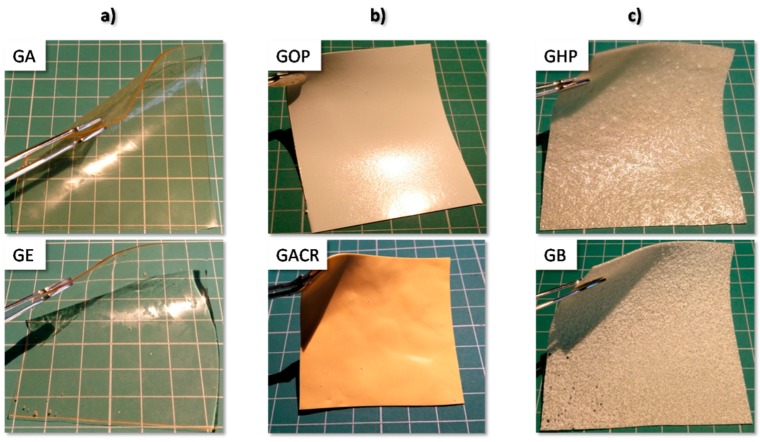
Appearance of example films containing various acid-resistant additives: (**a**) homogeneous and transparent; (**b**) apparently homogenous, colored; (**c**) with visible particles.

**Figure 3 polymers-11-00338-f003:**
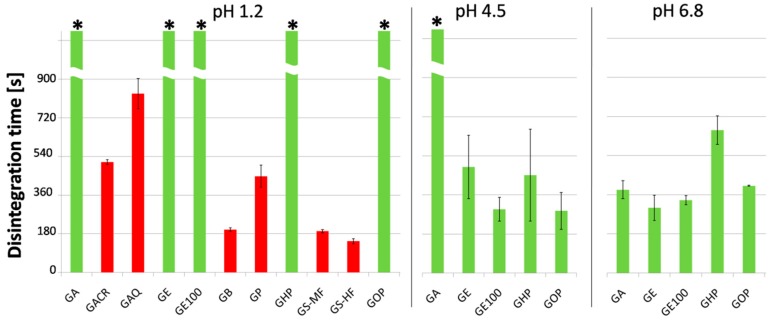
Disintegration time of investigated films (mean ± SD, *n* = 3). Red bars indicate compositions that failed the test at acid phase. (*)—disintegration not observed within 3 h.

**Figure 4 polymers-11-00338-f004:**
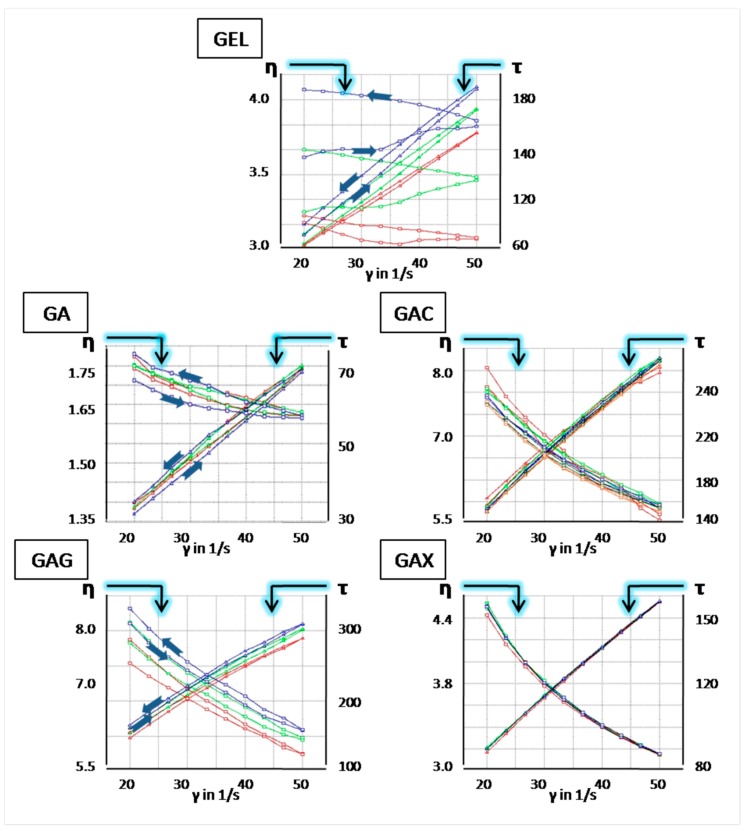
Graphs displaying relation of shear stress (τ (Pa)) and dynamic viscosity (η (Pa × s)) with shear rate (γ), measured for the film-forming mass at 80 °C. The effect of cellulose acetate phthalate and additional polysaccharides in the gelatin mass is presented. Measurements were performed in triplicate for the same film-forming mass sample: 1st cycle—red curve, 2nd—green, 3rd—blue. On the blue graphs (3rd measurement) with considerable hysteresis loops, arrows indicate the up-curves and down-curves.

**Figure 5 polymers-11-00338-f005:**
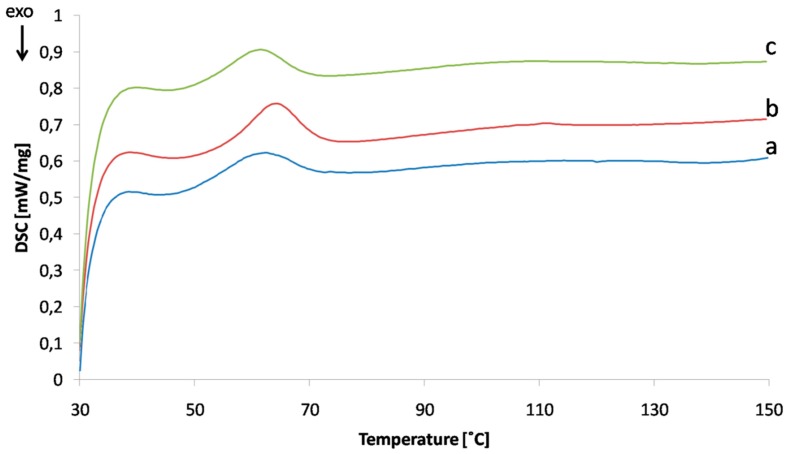
DSC curves of GA films ((**a**) after preparation; (**b**) after 10 months of storage) and GAC films (**c**).

**Figure 6 polymers-11-00338-f006:**
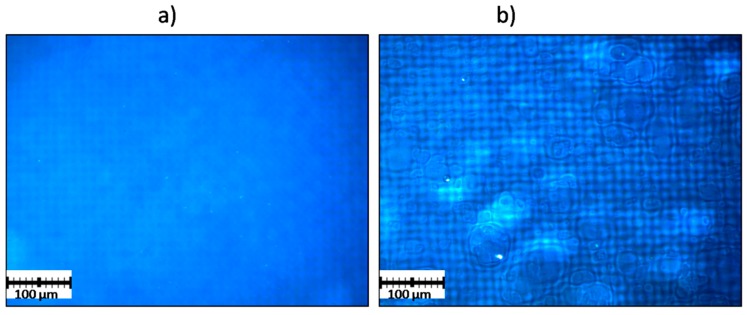
Film surface: (**a**) GAC; (**b**) GAG. The blue background with the paned pattern was used to better visualize the film structure due to the transparency of the samples. Scale bar—100 µm.

**Figure 7 polymers-11-00338-f007:**
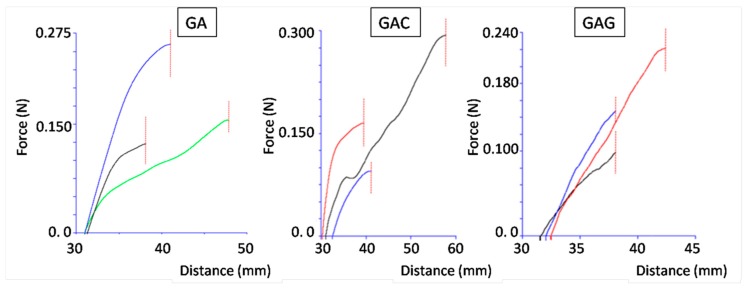
Results of adhesion tests for films fused at 60 °C (*n* = 3). The dashed line marks the tearing of films off-site of the fusion area (“partition point”).

**Table 1 polymers-11-00338-t001:** Qualitative compositions of prepared gelatin-based films for the purpose of compatibility evaluation, disintegration test, transparency, and differential scanning calorimetry (DSC). All compositions were plasticized with glycerol (0.46 g/g polymers).

Symbol	Acid-Resistant Ingredients	Comments	Appearance/Structure	Disintegration (+/-)		
				pH 1.2	pH 4.5	pH 6.8
GEL	-	Reference non-modified film	Clear, transparent	+	+	+
GA	Aquacoat CPD (Cellulose acetate phthalate (CAP))	Additives in Aquacoat CPD dispersion:poloxamer	Clear, slightly turbid, transparent	-	-	+
GE	Eudragit L30 D-55 (methacrylic acid copolymer)	Additives in Eudragit L30 D-55 dispersion: SLS, Polysorbate 80	Clear, transparent	-	+	+
GE100	Eudragit L100-55 (methacrylic acid copolymer)	Additives in Eudragit L100 powder: SLS, Polysorbate 80	Clear, transparent	-	+	+
GACR	Acryl Eze II (methacrylic acid copolymer)	Additives in Acryl Eze II powder: SLS, talc, TiO_2_ poloxamer, calcium silicate, sodium carbonate	Homogeneous, colored, opalescent	+	n/a	n/a
GAQ	Aquarius Control ENA (methacrylic acid copolymer)	Additives in Aquarius Control ENA powder: TEC, talc, TiO_2_, colloidal silica	Homogeneous, colored, opalescent	+	n/a	n/a
GOP	Opadry Enteric (Polyvinyl acetate phthalate (PVAP))	Additives in Opadry Enteric powder: TiO_2_, TEC, stearic acid	Homogeneous, colored, opalescent	-	+	+
GHP	Hypromellose phthalate (HPMCP)	Powder, low solution viscosity, *T*_g_ ca. 145 °C	Heterogeneous, opalescent, particles present	-	+	+
GB	Cellulose acetate butyrate (CAB)	Powder, medium solution viscosity, *T*_g_ ca. 130 °C	Heterogeneous, opalescent, particles present	+	n/a	n/a
GP	Cellulose acetate propionate (CP)	Powder, medium solution viscosity, *T*_g_ ca. 142 °C	Heterogeneous, opalescent, particles present	+	n/a	n/a
GS-MF GS-HF	Hypromellose acetate succinate (HPMCAS)	Powder, two grades tested: medium-acetyl (MF) and high-acetyl (HF), *T*_g_ 130–135 °C	Heterogeneous, opalescent, particles present	+	n/a	n/a

n/a—data not available. SLS—sodium laurylsulphate, TEC—triethyl citrate, TiO_2_—titanium dioxide, (+)—disintegration observed, (-)—disintegration not observed.

**Table 2 polymers-11-00338-t002:** GA compositions (gelatin with cellulose acetate phthalate), additionally modified with polysaccharides, added in the amount of 1.5% of the total dry content. All compositions were plasticized with glycerol.

Symbol	Gelling System	Acid-Resistant Ingredients	Appearance/Structure	Disintegration (+/-)		
				pH 1.2	pH 4.5	pH 6.8
GAC	Gelatin/ί-carrageenan	Aquacoat CPD (CAP)	Homogeneous, slightly turbid, transparent	-	-	+
GAG	Gelatin/gellan	Aquacoat CPD (CAP)	Slightly turbid, graininess visible in the structure	-	-	+
GAX	Gelatin/xanthan	Aquacoat CPD (CAP)	Slightly turbid, homogeneous, transparent	-	-	+

(+)—disintegration observed, (-)—disintegration not observed.

**Table 3 polymers-11-00338-t003:** Results of transparency measurement of GEL, GA, and GAC compositions.

Film Composition	Transmittance (600 nm)	Mean Thickness (mm)	Transparency
GEL	0.8702	0.495	−0.122
GA	0.8230	0.500	−0.169
GAC	0.6765	0.415	−0.409

**Table 4 polymers-11-00338-t004:** Mechanical parameters of casted films.

Film Composition	Moisture Content (% *w*/*w*)	Thickness (µm)	Force at Break (N)	Tensile Strength (kPa)	Elongation at Break (%)	Young’s Modulus (MPa)
GEL	2.3 ± 0.5	788 ± 44.6	>50 *	n/a *	n/a *	310.7 ± 61.0
GA	2.8 ± 0.2	639 ± 17.7	23.5 ± 2.8	7.36 ± 0.87	200% ± 10%	98.6 ± 16.5
GAC	2.8 ± 0.4	697 ± 33.0	20.2 ± 1.1	5.80 ± 0.32	226% ± 8%	51.4 ± 5.2
GAG	4.9 ± 0.2	704 ± 30.4	14.7 ± 1.3	4.16 ± 0.38	211% ± 7%	45.1 ± 8.3

* Force needed to disrupt the sample exceeded the maximum load of the equipment (>50 N).

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
