# Peer review of "Gelatin Films Modified with Acidic and Polyelectrolyte Polymers—Material Selection for Soft Gastroresistant Capsules"

_polymers, 2019, doi:10.3390/polym11020338_

Round 1
Reviewer 1 Report
The manuscript entitled ‘ Gelatin films modified with acidic and 2polyelectrolyte polymers – selection of material for 3 soft gastro-resistant capsules’by Maciejewski et al. discuss the development of acid resistant gelatin films for successful delivery of drugs in eneteric environment. Although, this is an interesting topic, which has seen much interest recently, the manuscript presented here does not meet the high quality of publications accepted is ‘Polymers’. In general, manuscript is poorly written and presented, is very hard to follow and misses important details on experimental procedures and discussion. Specifically,
1. Methods and experimental section need significant improvement. The authors should include detailed description of film synthesis and of parameters used for characterization. With the information provided in the manuscript, this is not possible to replicate any of these experiments.
For example: what were the conditions used for thermal analysis of samples, heat flow/cooling rate?
Explain in detail how disintegration test was performed?
For the analysis of tensile strength, what was the force applied to the films?
What molar or weight ratio of gelatin to copolymers was used for successful film synthesis? This will be helpful to incorporate the quantitative ratios of components (polymers, gelatin, plasticizers) required for synthesis of films in tabulated form.
2. The figures are presented in a very poor format and are unacceptable for the publication in good impact journal
3. The authors should look for a quantitative method to measure disintegration of polymer films, visual inspection does not qualify for quantitative measurement and are generally not reproducible.
4. This is unclear what is any is the impact of molar ratios of different polymers on gelatin films. Why GAC films were made at 100:3 ratios, were higher ratios tested? If not why?
5. DSC data obtained by authors is inconclusive. Why DSC test was performed on stored films? The authors indicate that there was a shift in temperature observed for stored GA films from 62.4 to 64 C. What does this shift indicate? The authors claim that functional properties of films were not impaired after storage, how this was measured?
In general, the manuscript does not provide significant contribution in terms of experimental details or results in this field of science and cannot be accepted for publication.
Author Response
Reviewer 1.
1. Methods and experimental section need significant improvement. The authors should include detailed description of film synthesis and of parameters used for characterization. With the information provided in the manuscript, this is not possible to replicate any of these experiments.
The description of film preparation has been elaborated in section 2.2, line 104, by rewriting the paragraph as follows:
“All film compositions contained gelatin, acid-resistant additive, and plasticizer. The mass ratio of the acid-resistant component to gelatin was 1:3. The compositions contained glycerol as a plasticizer, in an amount of 0.46 per 1 g of total polymer amount. The first step of film preparation was mixing all components with water. The initial water content in all compositions was 40% w/w. The mixture was heated in a water bath to 80˚C while being stirred with a paddle stirrer at 40 rpm for 2 h. Afterward, it was de-aerated under vacuum. The hot film-forming mass was then casted on a glass plate, and the thickness of the film was aligned by sliding through a 2000 µm gap, using a Camag TLC plate coater (Camag, Muttenz, Switzerland). Afterward, the films were dried in an air dryer at room temperature for 90 min and finally equilibrated at room temperature/15%–25% RH for at least 24 h. This procedure resulted in dry films with 450–700 µm thickness.”
a) What were the conditions used for thermal analysis of samples, heat flow/cooling rate?
The DSC measurement protocol was indeed originally lacking in the text. The conditions have been described in the paragraph 2.8., line 190:
“The film samples (5 mg) were placed in an aluminum pan and heated in a nitrogen atmosphere from 30 to 150 ˚C with a rate of 10 K/min in a single cycle (without a cooling phase), with the empty aluminum pans as a reference.”
b) Explain in detail how disintegration test was performed?
The disintegration test utilized a standard apparatus and method described in the European Pharmacopeia, therefore, initially the detailed description was skipped. According to Reviewer’s suggestion, the method has been described in the section 2.3., line 141, as follows:
“Disintegration of the films was evaluated using a pharmacopoeial (European Pharmacopoeia, Ph. Eur.) tablet disintegration tester (ED 2SAPOx, Electrolab, Mumbai, India). Film samples 1×1 cm were placed in steel mesh sinkers before being introduced to the apparatus baskets. The test was performed at 37˚C using Ph. Eur. media (900 ml) at pH 1.2 (Simulated Gastric Fluid, SGF), pH 4.5 (phosphate buffer), and pH 6.8 (phosphate buffer). The baskets with the samples were repeatedly raised and lowered in the immersion fluid at a frequency rate of 30 cycles per min. The endpoint of the test was defined when any discontinuities in the film sample were observed.”
c) For the analysis of tensile strength, what was the force applied to the films?
The measurement was performed using a texture analyzer with a 5 kg load cell, what have been explained in section 2.6.The values of force measured at rupture of the films have been added to the table 3. in section 3.4.1. (line 353).
d) What molar or weight ratio of gelatin to copolymers was used for successful film synthesis? This will be helpful to incorporate the quantitative rations of components (polymers, gelatin, plasticizers) required for synthesis of films in tabulated form.
The weight ratio of gelatin to enteric polymers was 3:1. This was indicated in the section 2.2., line 105. All films had similar composition, resulting from keeping the same ratio of gelatin to acid-insoluble additive, the same plasticizer amount and the same initial water content. In the case of added compound enteric mixtures, i.e. Acryl Eze II, the quantitative composition is unavailable due to mixture producer’s policy. Therefore, showing the quantitative compositions of film formulation in a tabulated form will not provide any new information.
2. The figures are presented in a very poor format and are unacceptable for the publication in good impact journal
The quality of figures has been improved in terms of size and DPI values.
3. The authors should look for a quantitative method to measure disintegration of polymer films, visual inspection does not qualify for quantitative measurement and are generally not reproducible.
The disintegration tests that are used as a routine assessment of functionality of drug dosage forms in pharmaceutical technology are generally qualitative tests, where endpoints are indicated by visual inspection. Such tests can be skipped and replaced by subjecting a pharmaceutical dosage form to a dissolution test, in which samples of acceptor medium are tested at certain time points and concentration of active substance that was dissolved is measured. However, in case of this study, the intention was to assess the potential feasibility of prepared membranes to further use as an enteric capsule shell. At the present stage of work, we are testing whether the prepared films remain their structural integrity when they are submersed in acidic medium. In such kind of test, a visual observation of membranes is needed.
It is important to notice that most of the film disintegration measurement methods described in the literature are based on the same principle, which is visual endpoint determination, and utilize either pharmacopoeial tablet disintegration tester, or simple glass vessel filled with disintegration medium. The differences between the methods are based on construction of a sample holder (i.a. Speer, I. et al. Eur. J. Pharm. Biopharm. 2018, 132, 50–61; Low, A. et al. J. Pharm. Sci. 2015, 104, 3893–3903). An exception can be a method, in which a loaded clip is attached to a vertically placed film sample, which is placed in a specially designed sample holder of a modified tablet disintegration tester (Preis, M. et al. J. Pharm. Pharmacol. 2014, 66, 1102–1111). Such method, however, requires a full process development regarding testing conditions and weight that can be applied to the film sample. It is also difficult to predict, what the correlation between the results obtained with the developed test and actual performance of prepared enteric capsules will be.
The trials to develop a qualitative method for assessing the disintegration time were described in the literature in the case of ODFs (Oral Thin Films) disintegration time measurement. The mentioned method comprised a specially designed cell, in which a small volume of fluid (20-40 ml) is placed on top of the investigated film sample, and the disintegration time is detected automatically by an optical passage confirmation sensor that is placed below the film sample and is activated by a drop of fluid that penetrated the sample (Takeuchi, Y. et al. Int. J. Pharm. 2018, 553, 281–289). Such small volume is never the case in the gastric environment, and can be utilized only in orally disintegrating forms. Moreover, this method requires specially designed equipment, which could not be accessed during our study.
It is worth to mention, that our previous work comprised a permeability test, in which a permeation rate of a radio-labeled water through various membranes at pH 1.2 was measured (Maciejewski, B. et al, Polymers 2018, 10, 981). However, such method is expensive and very hard to utilize at regular basis in the first trials of novel film materials comprising various acid-insoluble polymers. Therefore, a screening investigation with use of disintegration test with visual endpoint can be considered beneficial.
The further development of the described materials will comprise a preparation of liquid-filled capsules with active ingredients. Only at that stage it will be possible to perform a dissolution test that will provide information about an actual barrier property of developed material against acid.
4. This is unclear what is any of the impact of molar ratios of different polymers on gelatin films. Why GAC films were made at 100:3 ratios, were higher ratios tested? If not why?
Our preliminary studies showed that only with gelatin in the concentration around 30 % and 3:1 gelatin to enteric polymer ratio, the films with proper disintegration, elasticity and setting properties can be prepared. This was evaluated and published in our previous work (Maciejewski, B. et al. Pharmazie 2017, 72, 324–328; Maciejewski, B. et al. Polymers 2018, 10, 981). Similar results were obtained in preliminary tests for films modified with addition of Eudragit L. The addition of other polymers and coating mixtures was not investigated at that stage of work.
Polysaccharides utilized in this study form very viscous gels at low concentrations in water (ca. 2%). It was confirmed that at higher ratios of polysaccharides to gelatin, the film-forming solutions were hard to deareate and to cast due to very high viscosity. Moreover, the obtained films were characterized by a very small mechanical resistance. Finally, only at the mass ratio of gelatin to polysaccharide around 100:3 the films exhibited optimal processability and could be further analyzed.
5. DSC data obtained by authors is inconclusive. Why DSC test was performed on stored films? The authors indicate that there was a shift in temperature observed for stored GA films from 62.4 to 64 C. What does this shift indicate? The authors claim that functional properties of films were not impaired after storage, how this was measured?
The DSC test was performed on stored films to investigate possible changes in thermal effects that could be related to any reactions in film material on storage that can occur due to relatively low stability of gelatin-based formulations. The factors impairing the stability of said compositions can be either structural water content (leading to potential hydrolysis or microbiological instability), or tendency of gelatin to form crosslinks between its chains. The crosslinking is a common issue in gelatin-based oral formulations, because it leads to formation of material insoluble regardless of pH of the acceptor fluid. Not knowing the exact mechanism of formation of modified gelatin-CAP films, it could be possible that CAP induces the crosslinking reaction in the gelatin domains, which would lead to impairing the functionality of capsules prepared with such material.
The shift in temperature of an endothermic peak can be connected either with melting processes, or evaporation. Theoretically, the gelatin consists of both amorphous and crystalline domains. The amount of crystalline regions can potentially vary on storage, which could be an explanation to the temperature shift. However, the XRD analysis did not confirm presence of crystalline structures in the GA films, neither stored nor freshly prepared (results not shown in the article). The XRD results are shown in the figure below (red graph stands for fresh GA sample, green graph for sample stored for 10 months.
Figure 1. XRD graphs of film samples. Red graph – GA, Green graph – GA stored for 10 months, Blue graph – GAC, Violet graph – reference GEL sample.
Therefore the shift in temperatures can be connected either to evaporation of volatile ingredients of the sample, or can be effect of a sample-to-sample variation. However at this stage, it cannot be concluded whether the obtained results are significant or not. Anyhow, the disintegration characteristics of the films remained unchanged. This was measured with disintegration test according to protocol described in the manuscript (explanation has been added in section 3.3., line 323, as follows:
“functional properties of films (resistance to acid, while fast dissolution at pH 6.8, evaluated with disintegration test) were not impaired”,
however we decided to not include the results in the paper in order to avoid creating confusion.

Reviewer 2 Report
SEM analysis can help observe gel microstructure and supramolecular assemblies.
Author Response
No comments have been received at the second run

Reviewer 3 Report
The authors prepared the gelatin films containing cellulose derivatives and methacrylic acid copolymers for gastro-resistant capsules. Some information and explanation are still lacked. In my opinion, the authors should major revision the manuscript in order to “Polymers” because of the following comments.
1. Line 50, the authors mentioned, “it is still unclear how important are chemical interactions between functional groups of gelatin’s amino acids and the groups in added polymers”. Line 403, the authors mentioned, “No interaction of chemical nature was detected between gelatin and additive polymers”. The results of the manuscript can’t convince me to realize the conclusion.
2. Line 88, “The film-forming masses were prepared by mixing the ingredients in a heated round flask at 80˚C for 2 h. The total water content in all compositions was 40%w/w. The mass ratio of the acid-resistant components to gelatin was 3:1.”. Please explain why all formulations use the same synthesized conditions and cellulose derivatives and methacrylic acid copolymers were used as raw materials. Especially, solubility parameters of some raw materials were totally different.
3. Table 1, please explain what is + and -. This Table let me very confused.
4. Figure 1 and 6, Please refer other studies to show what kind of appearance and transparency is suit to oral delivery.
5. Figure 2, how long is disintegration time enough to deal with gastro-resistant capsules.
6. Figure 4 is not clear.
7. Figure 5, DSC analyses should be showed on the curve and the figure is not clear. Why other samples did not measure?
8. Table 2, what is unit of transmittance? Line 133 mentioned, “The absorbance was measured at 650, 600, 570, 530, 510, 470, 440 and 400 nm wavelengths, representing 7 basic colors.” There are only 600 nm results in the manuscript.
9. Table 3, Please revise the unit of moisture content.
10. Line 344 “3.5. Summary” should be merged and rewritten to the other section or revise to the discussion.
11. Please revise “5” Conclusions to “4” Conclusions. The authors should tell the reader which formulation is optimized.
12. There are many typographical and grammatical errors in the manuscript.
Author Response
Reviewer 3.
1. Line 50, the authors mentioned, “it is still unclear how important are chemical interactions between functional groups of gelatin’s amino acids and the groups in added polymers”. Line 403, the authors mentioned, “No interaction of chemical nature was detected between gelatin and additive polymers”. The results of the manuscript can’t convince me to realize the conclusion.
We agree with the Reviewer that the conclusion was not fully supported with the results and design of the study. The sentence was removed from the conclusion section and following sentence was added (line 459):
“No relation was observed between the chemical nature of the enteric components and their ability to form with gelatin acid-resistant films, but the formation of such films was most likely caused by the uninterrupted coalescence of dispersed polymer particles, resulting in the films’ homogeneous morphology. Interestingly, this process depended on the additives present in the commercial mixtures of the enteric polymers, and some of these products were unsuitable despite the same type of the present enteric polymers”.
2. Line 88, “The film-forming masses were prepared by mixing the ingredients in a heated round flask at 80˚C for 2 h. The total water content in all compositions was 40%w/w. The mass ratio of the acid-resistant components to gelatin was 3:1.”. Please explain why all formulations use the same synthesized conditions and cellulose derivatives and methacrylic acid copolymers were used as raw materials. Especially, solubility parameters of some raw materials were totally different.
The intention behind the presented study was to assess the compatibility of gelatin and acid-insoluble ingredients under conditions, which were optimized during preliminary experiments. The key conditions of the film-forming mass preparation process cannot be changed, due to properties of the materials used in the compositions. As an explanation: higher mixing rates create higher foaming, and makes deareation difficult; higher temperature cannot be used due to risk of gelatin hydrolysis, while lower temperature is not beneficial in terms of polymer chain mobility and viscosity of the prepared film-forming mass; finally, longer process time creates risk of higher water loss, while shorter time is not always assuring uniform dissolution of soluble components. The composition of the film-forming masses was also optimized in terms of possible usefulness in soft capsule technology. For this purpose, it was noted that the lower initial water content and the higher gelatin concentration, the better setting properties of the resulting casted films. Moreover, it was noted that to render enteric properties of the resulting films possible, without impairing the mechanical characteristics of the formulation, the ratio of acid-resistant component to gelatin should be not more than 1:3. However we realize that the optimal parameters for certain compositions can slightly differ; the main purpose of this specific work was to evaluate possibilities of formation of stable films with use not only of raw materials, but also of commercial dispersions and mixtures used in pharmaceutical technology for enteric coating. Therefore, the experiments were planned only to allow selection of the most promising compositions for further optimization and development.
3. Table 1, please explain what is + and -. This Table let me very confused.
In Table 1, (+) means that disintegration of certain film composition was observed under investigated conditions, while (-) is used to mark a result, where certain film composition did not disintegrate under investigated conditions. The explanation has been added as footnotes to the tables 1 and 2. in section 2.2., lines 122 and 128.
4. Figure 1 and 6, Please refer other studies to show what kind of appearance and transparency is suit to oral delivery.
In oral delivery, appearance and transparency is not a determining factor in terms of functionality, therefore we have not found in a literature any overarching requirements to refer to. The aimed appearance can be dependent mainly on manufacturer’s specification or preferences of a patient. In terms of acceptance of a dosage form by patient, it can be assumed that solid oral dosage forms with smooth surface are easier to swallow than rough ones. Therefore, striving for smoothness of a surface can be reasonable.
The transparency can be related to barrier properties of a material against light, and ease of manipulation of colour by manufacturer at further stages of product development. In case of the following work, the transparency parameter was used as an indicator of phase separation within the film material, i.e. the more cloudy film is, the more particles or separate domains can be expected in the material. Additionally, in the case of soft gelatin capsules, usually it is desired to form a clear and transparent capsule, due to ease of modification of appearance of such compositions. Intense colour or opalescence of capsule shell material can make the appearance of a product hard to design.
5. Figure 2, how long is disintegration time enough to deal with gastro-resistant capsules.
The current standards for disintegration time of gastroresistant dosage forms state that the investigated sample should not disintegrate in 0.1 M HCl within 2 or 3 h (depending on the composition, however not less than 1 h), which should be followed by disintegration within 1 h at pH 6.8 (European Pharmacopeia 9.0). The description of a disintegration time standard has been added to the text in section 3.1., line 236.
6. Figure 4 is not clear.
To make the graphs clearer, additional information has been added to the caption under the figure 4., line 297:
“Graphs displaying relation of shear stress (τ [Pa]) and dynamic viscosity (η [Pa×s]) with shear rate (γ), measured for the film-forming mass at temperature of 80°C. The effect of cellulose acetate phthalate and additional polysaccharides in the gelatin mass is presented. Measurements were performed in triplicates for the same sample of the film-forming mass.”
7. Figure 5, DSC analyses should be showed on the curve and the figure is not clear. Why other samples did not measure?
Figure 5 was indeed not very clear, due to formatting issues. This figure has been replaced with combined thermograms of all investigated film samples, and the quality of figure has been improved.
We studied only GA films with DSC as they were selected as most suitable for further development due to the best disintegration parameters and appearance. On the other hand after modification with polysaccharide, the obtained GAC films (additionally containing carrageenan) were the only fully homogenous composition, displaying altered properties such as rheology, tensile strength and adhesiveness. Therefore DSC information was the most desired in case of these samples. The purpose of this DSC study was to investigate formation of a polyelectrolyte complex between gelatin and carrageenan. The second purpose was to compare the thermograms of freshly prepared GA samples with samples stored for 10 months, in order to find traces of degradation processes or gelatin crosslinking that could have effect on the thermal characteristics of the sample. We have assumed that a thorough investigation of all the obtained acid-resistant films was not required at this stage of development.
8. Table 2, what is unit of transmittance? Line 133 mentioned, “The absorbance was measured at 650, 600, 570, 530, 510, 470, 440 and 400 nm wavelengths, representing 7 basic colors.” There are only 600 nm results in the manuscript.
The light transmittance in table 2 was measured as a function of absorbance, according to Beer-Lambert law, therefore it is expressed without a unit.
The sentence in the line 133 was an artifact from early stages of manuscript preparation, and was kept in the text by mistake. Indeed, the results shown and discussed in the following work contain only transmittance at 600 nm wavelength. Therefore, the sentence mentioned above has been removed.
9. Table 3, Please revise the unit of moisture content.
The unit of moisture content in the table 3, line 353, has been corrected to “% w/w”.
10. Line 344 “3.5. Summary” should be merged and rewritten to the other section or revise to the discussion.
The summary section had been introduced following the rules already practiced in some of the articles published in the journal. As suggested by the Reviewer we have replaced this by Discussion section, which was improved by adding some new comments (lines 387-391, 405-411, 448-454).
11. Please revise “5” Conclusions to “4” Conclusions. The authors should tell the reader which formulation is optimized.
Originally, the heading of “Conclusions” section was given a number 5 by mistake. However, to address the Reviewer’s suggestion no. 10, the construction of the manuscript has been changed by adding the “Discussion” section, which has been given number 4. Therefore, the “Conclusions” should remain with number 5.
The sentence containing selection of the best optimized formulation has been added to the “Conclusions” section, as follows:
“Anyhow, due to certain issues with film homogeneity regarding compositions with gellan (GAG), the composition with carrageenan (GAC) appears to be the most suitable for further development.”
12. There are many typographical and grammatical errors in the manuscript.
Thank you for this comment. The manuscript has been revised by using the editing help provided by the Journal and the grammar errors have been corrected.

Round 2
Reviewer 1 Report
The manuscript entitled ' Gelatin Films Modified with Acidic and Polyelectrolyte Polymers—Material Selection for Soft Gastroresistant Capsules' shows significant improvement compared to previously submitted versions. However, i do have following concerns:
1. Figure 4, which has no legend.
2. Spelling and grammar needs careful revision, example, “thixotropic” is written as “tixotropic”.
3. The section on Rheology is very unclear and the data obtained is not very well explained. The authors are encouraged to discuss the rheology of materials in detail.
Author Response
Reviewer 1.
The manuscript entitled ' Gelatin Films Modified with Acidic and Polyelectrolyte Polymers—Material Selection for Soft Gastroresistant Capsules' shows significant improvement compared to previously submitted versions. However, I do have following concerns:
1. Figure 4, which has no legend.
Each graph in the figure 4. shows 3 replicates of the measurement performed on the same sample of certain composition. Due to large amount of data points, we decided to use colours to differentiate subsequent measurements of the sample instead of incorporating legend in the figure. However, we do understand that description of the data on the graphs required improvement. Therefore, we propose an explanation of the replicates in the figure caption, as follows (line 284):
“Graphs displaying relation of shear stress (τ (Pa)) and dynamic viscosity (η (Pa×s)) with shear rate (γ), measured for the film-forming mass at 80°C. The effect of cellulose acetate phthalate and additional polysaccharides in the gelatin mass is presented. Measurements were performed in triplicate for the same film-forming mass sample (1st red, 2nd green, 3rd blue). On the blue graphs (3rd measurement) with considerable hysteresis loops arrows indicate the up-curves and down-curves.”
Moreover, the explanation has been added in the text in line 286.
Additionally we changed the attribution of curves to the axes of the graphs in an attempt to present it more clearly.
2. Spelling and grammar needs careful revision, example, “thixotropic” is written as “tixotropic”.
The text was subjected to English editing using an editing service provided by MDPI. The certificate has been uploaded along with the previous version of the manuscript. However, “thixotropic” was spelled wrong indeed due to an oversight, and we apologize for that. As the text has been changed this word does not appear in the actual version.
3. The section on Rheology is very unclear and the data obtained is not very well explained. The authors are encouraged to discuss the rheology of materials in detail.
The description of obtained results has been elaborated according to Reviewer’s suggestion. The text in section 3.2. has been edited, gaining the following form (lines 259 - 332):
“In the next step of the study, gastroresistant GA was modified by the addition of a secondary gelling agent such as carrageenan, gellan, and xanthan, as described in the Methods section (Table 2.). In this type of polymers, one can expect that polyelectrolyte complexes with gelatin are formed [20–22], which may change some of the physical properties of the films.
Figure 4. shows results of rheological measurements of 5 investigated film-forming mixtures: GEL (as reference), GA, GAC, GAG, and GAX. The compositions were investigated under isothermal conditions at 80˚C, what reflects the temperature of the gelatin mass prior to film casting during soft capsules manufacturing. Thermostated hot mixtures were investigated with a cylinder probe according to the test protocol described in the Methods section. Three subsequent measurements of the same sample were performed within 30 min (each run lasted 10 min). Such design of the experiment was aimed to better differentiate rheological properties of individual compositions and to detect changes caused by additives. In Figure 4 red curves refer to 1st run, green and blue for 2nd and 3rd replicate, respectively.
There is a large difference between viscosity values measured for GEL and GA formulations. It is demonstrated that the viscosity of gelatin mass is much lower (more than 2 times) upon addition of CAP (GA composition), which can cause trouble in the industrial production of soft gelatin shells, because too small viscosity of the casted mass creates risk of inability of forming films with uniform and proper thickness. However, a small content of cogelling polymers in polysaccharide-modified mixtures (GAC, GAX, GAG) resulted in significant increase of the viscosity which was similar to the viscosity of GEL mass or even higher. This can be considered an advantage in terms of handling of such film-forming masses.
The analysis of the viscosity and shear stress curves generated during three subsequent measurements also demonstrates significant difference between GEL and GA formulations. A very pronounced dilatant behavior of the GEL mass is observed, which can be probably explained by easy gelation of the mass in contact with air during mixing, when water evaporation from the hot mass also occurs. Moreover, in the GA and GAG modified mass dilatancy is still visible, however to much lesser extent. When carrageen or xanthan gum is added to GA the mixtures are physically very stable and no changes in the curves generated in the subsequent 3 runs were observed (GAC and GAX). Therefore, besides other advantages of gelling system modification, addition of cogelling agents to the formulation could result in better processability of GA-based compositions.
The obtained results suggest a stabilizing effect of a polysaccharide on GA mass, which suspectedly is based on an ionic interaction. According to the literature [3], the isoelectric point (pI) of gelatin is about 9.0. Xanthan gum is a polysaccharide polymer that can form complex hydrogels with gelatin based on electrostatic interaction when the pH of the mixture is around or below the pI of gelatin, and the temperature is above the coil–helix transitions of both materials [21]. Similarly, gellan and carrageenan are anionic polysaccharides that form mixed gel networks, stabilized by the electrostatic attraction of molecules [22,23]. Regardless of the polymer and other additives used, the pH was, in all compositions, in the range of 4–5, which was below the pI of gelatin. Since the ratio of gelatin and these additional polymers was high (ca. 34:1), the system was still rich in ionized functional groups originating from gelatin.”
Moreover, an information about the sample volume (40 ml) has been added to the section 2.4. (line 135).
Reviewer 3 Report
My comments have been modified accordingly. This article could be accepted in present form.
Author Response
The text was subjected to English editing using an editing service provided by MDPI. The certificate has been uploaded along with the previous version of the manuscript.
Round 3
Reviewer 1 Report
The authors have addressed the relevant comments. I would suggest accepting this paper after careful revision of English language.